# Learning Probabilistic Logic Models over Structured and Unstructured Data

## Abstract

Effective decision-making in high-stakes domains necessitates reconciling information from structured and unstructured data with incomplete and imprecise background knowledge. Relational Dependency Networks are a popular class of probabilistic logic models that support efficient reasoning over structured data and symbolic domain knowledge but struggle to accommodate unstructured data such as images and text. On the other hand, neural networks excel at extracting patterns from unstructured data but are not amenable to reasoning. We propose Deep Relational Dependency Networks which combine Relational Dependency Networks with neural networks to reason effectively about multimodal data and symbolic domain knowledge. Experiments on scene classification tasks with noisy and limited data indicate that this approach yields more accurate yet interpretable models.

## Introduction

The exponential growth and heterogeneity of available data present significant challenges for decision-making, particularly in high-stakes domains such as healthcare (Norman et al. 2017). Effective clinical reasoning often requires integrating and interpreting information from structured electronic health records (EHRs) and unstructured sources like medical imaging data and free-text notes (Chin-Yee and Upshur 2018). Clinicians must also reconcile this information with decades of accumulated domain knowledge—encompassing medical research findings, clinical guidelines, and expert heuristics—into their diagnostic, prognostic, and treatment decisions. Decision support systems can help reduce the decision-maker's cognitive load by partially automating this process (Rabaey et al. 2024).

Probabilistic Logic Models (PLMs) (Getoor and Taskar 2007), such as Relational Dependency Networks (RDNs) (Neville and Jensen 2007), offer a powerful framework for reasoning under uncertainty in such complex, inherently multi-relational domains. RDNs exploit relational symmetries to efficiently model complex, multi-entity interactions and integrate symbolic domain knowledge. This allows them to produce more interpretable and robust inferences that align with expert-validated principles. However, the practical deployment of RDNs in real-world decision support remains limited by their reliance on fully structured data representations. In realistic environments, critical informa-

tion often resides in unstructured modalities—such as medical scans, pathology images, or textual clinical notes—and transcribing these raw data into structured form introduces information loss and additional engineering overhead.

In this work, we address these limitations by extending RDNs to directly accommodate unstructured inputs. We achieve this by coupling RDNs with neural network representations that extract semantic features from raw modalities. By integrating these learned embeddings into the relational inference process, our approach preserves the relational reasoning capabilities of PLMs while harnessing the representational power of deep learning. This results in a more flexible, information-rich modeling framework that can handle the complexity and diversity of modern data ecosystems in high-stakes decision-making domains.

## Background

### (Relational) Dependency Networks

Dependency Networks (DNs) (Heckerman et al. 2000) are directed probabilistic graphical models (Koller 2009) that approximate a joint probability distribution over a set of variables using a collection of conditional distributions, one per variable. Unlike Bayesian Networks (BNs), which impose acyclicity, DNs can represent cyclic dependencies, allowing them to capture more complex, real-world phenomena. Although this flexibility can sometimes lead to approximate or inconsistent joint distributions, DNs often provide computational advantages in learning and inference.

Formally, let $\mathbf{X} = \{X_i\}_{i=1}^n$ be a set of $n$ random variables, and $\{P_i(X_i \mid \mathbf{X}_{-i})\}_{i=1}^n$ be the set of local conditional distributions, where $\mathbf{X}_{-i}$ denotes all variables except $X_i$. A DN approximates the joint distribution as:

$$P(\mathbf{X}) \approx \prod_{i=1}^n P_i(X_i \mid \mathbf{X}_{-i})$$

Relational Dependency Networks (RDNs) (Neville and Jensen 2007) extend DNs to compactly represent joint probability distributions over complex, multi-relational domains by exploiting their structural symmetries. Just like their propositional counterpart, RDNs approximate the joint distribution over a relational domain as the product of local conditionals. However, the conditional for each relational logic predicate is shared across all of its groundings.

An RDN over a relational domain $\mathcal{D}$ consisting of relations $R$ and objects $O$ is defined using a set of structured local conditionals over groundings of individual relations given all ground atoms. Then, the joint distribution over a grounded database $\mathcal{A}(D, R) = \mathbf{a}$ can be approximated as:

$$P(\mathcal{A}(D, R) = \mathbf{a}) \approx \prod_{r \in R} \prod_{r(\mathbf{o}) \in \mathbf{a}} P_r(r(\mathbf{o}) \mid \mathbf{a} \setminus r(\mathbf{o}))$$

where each $r(\mathbf{o})$ is an instantiation of the relational predicate $r \in R$ with an object $o \in O$ in the grounded database $\mathbf{a}$, and each $P_r$ is the local conditional distribution for $r$.

These conditionals can be learned from data as Relational Probability Trees (RPTs) (Blockeel and Raedt 1998) and gradient-boosted Relational Regression Trees (GB-RRTs) (Natarajan et al. 2012).

## Domain Knowledge as label preference rules

While purely data-driven learning methods have achieved remarkable success (Mitchell 1997; Cristianini 2000; Schapire and Freund 2013), they often struggle when faced with noisy or incomplete observations. This challenge becomes more pronounced in relational domains, where the complexity of the data and the exponential growth in possible relationships can make data collection and labeling expensive and error-prone. In such situations, exploiting domain knowledge, often held by human experts, becomes invaluable. By providing insights and constraints that guide model construction (McCarthy 1959), domain knowledge can significantly enhance robustness and accuracy (Baffes and Mooney 1996), particularly when labeled data is noisy and limited (Yang and Natarajan 2013; Kokel et al. 2020; Karpatne, Kannan, and Kumar 2022; Mathur, Gogate, and Natarajan 2023; Mathur, Antonucci, and Natarajan 2024).

Label preference rules (Boutilier 2002; Odom and Natarajan 2018) are a powerful framework for representing such knowledge. Label preference rules express partial domain knowledge about the relationships between multi-relational data and target labels. These rules specify which labels are favored or disfavoured for specific subsets of examples. These rules can encode diverse forms of domain knowledge. For instance, knowledge about monotonicities (Altendorf, Restificar, and Dietterich 2005), synergies (Yang and Natarajan 2013), class-imbalance tradeoffs (Yang et al. 2014), and privileged information (Vapnik and Vashist 2009) can be represented as label preference rules.

Concretely, we can learn conditional distributions over a boolean target concept (say, $Y$) given a set of features (say, $\mathbf{X}$) from small and noisy data sets (say, $\mathcal{D}$) by exploiting domain knowledge in the form of positive and negative label preferences rules (say, $r_t(x)$ and $r_f(x)$) by minimizing a modified log likelihood-based objective function that includes a penalty term measuring the deviation of the model from the target. If the conditional is defined as $P(Y = 1 \mid \mathbf{X} = \mathbf{x}) = \sigma(\psi(\mathbf{x}))$ where $\sigma$ is the sigmoid function and $\psi$ is a real-valued potential function, the modified objective function is given by the following equation:

$$MLL(\psi, \mathcal{D}) = -\mathcal{L}(\psi, \mathcal{D}) - \lambda \sum_{(\mathbf{x}, y) \in \mathcal{D}} \psi(\mathbf{x})(n_t(\mathbf{x}) - n_f(\mathbf{x}))$$

where $n_t$ and $n_f$ are the number of preference rules that were true and false for $\mathbf{x}$ respectively. This augmented objective encourages the learned model to align with expert knowledge, leading to improved robustness and interpretability.

## Deep Relational Dependency Networks

We aim to learn probabilistic logic models (PLMs) from datasets that contain both structured (e.g., relational tables, knowledge graphs) and unstructured (e.g., images, text) data. The structured data ($Z$) provides a well-defined symbolic representation of entities and relationships, while the unstructured data ($X$) often conveys complementary, richer information that is not easily captured by relational predicates alone. Integrating these two modalities poses significant challenges, particularly when domain knowledge is available and must be leveraged, yet the data is incomplete or only partially observable. Existing frameworks for advice-based learning in PLMs primarily focus on scenarios where only structured data is observed and is sufficient for inferring the target labels. However, in multimodal settings, structured data might offer only partial information, requiring an adaptation of advice to handle partial observability. We first formalize the learning problem and then describe our proposed neurosymbolic framework, which we refer to as a *Deep Relational Dependency Network* (Deep-RDN).

**Problem Definition.** Our task is to learn a conditional distribution over a target concept ($y \in \{0, 1\}$) given data in structured ($z \in Z$) and unstructured ($x \in X$) forms from a dataset ($\mathcal{D}$) and symbolic domain knowledge ($\mathcal{K}$).

> **Given:** Data set $\mathcal{D} = \{(x^{(i)}, z^{(i)}, y^{(i)})\}_{i=1}^N$ and symbolic domain knowledge $\mathcal{K}$.
> **To Do:** Learn a deep probabilistic logic model $\mathcal{M}$ that accurately models $P(Y \mid X, Z)$.

This task can be formulated as the optimization problem

$$\arg\min_{\mathcal{M}} -\mathcal{L}(\mathcal{M}, \mathcal{D}) \text{ s.t. } \mathcal{M} \text{ satisfies } \mathcal{K}$$

where $-\mathcal{L}(\mathcal{M}, \mathcal{D})$ is the negative (conditional) logliklihood of the data set $\mathcal{D}$ under the distribution induced by the model $\mathcal{M}$. This is an exceedingly hard problem to solve in general. So, we solve it approximately by relaxing the hard constraint to a soft penalty. The resulting problem is

$$\arg\min_{\mathcal{M}} -\mathcal{L}(\mathcal{M}, \mathcal{D}) - \lambda \zeta(\mathcal{M}, \mathcal{K}, \mathcal{D}) \qquad (1)$$

where $\zeta(\mathcal{M}, \mathcal{K}, \mathcal{D})$ is a penalty function measuring the degree of violation of the domain knowledge by the model. We assume that $\mathcal{M}$ induces a conditional of the form

$$P(Y \mid X, Z) = \sigma(\psi(x, z))$$

where $\sigma$ is the sigmoid function and $\psi(x, z) \in \mathbb{R}$ is the potential function. Then, the penalty function $\zeta(\mathcal{M}, \mathcal{K}, \mathcal{D})$ for symbolic domain knowledge $\mathcal{K}$ in the form of preference rules can be defined as

$$\zeta(\mathcal{M}, \mathcal{K}, \mathcal{D}) = \sum_{(x, z, y) \in \mathcal{D}} \psi(x, z)(n_t(z) - n_f(z))$$

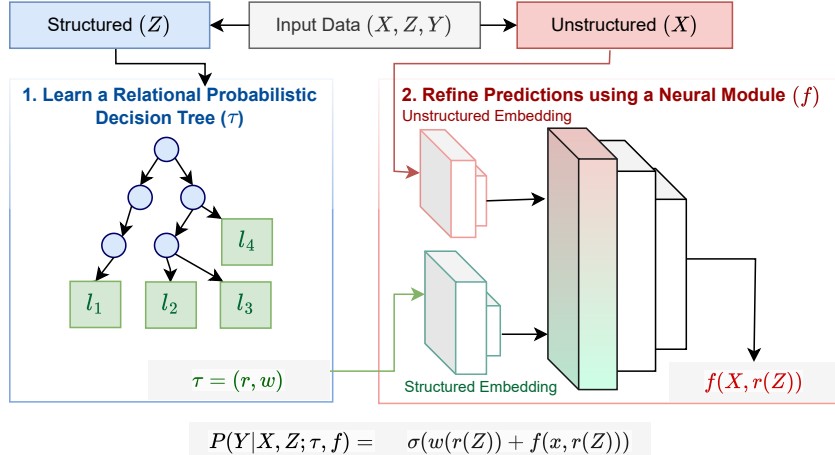

$$P(Y|X,Z;\tau,f) = \sigma(w(r(Z)) + f(x,r(Z)))$$

**Figure 1:** Overview of the Proposed Deep-RDN framework comprising two components: (1) A relational probabilistic decision tree ($\tau$) that partitions the structured data ($Z$) into related groups for predicting the target class ($Y$), and (2) A neural module that integrates rich unstructured data ($X$) with the tree's relational features to refine predictions. Together, these components collaboratively model the conditional probability $P(Y \mid X, Z; \tau, f)$, leveraging both symbolic reasoning and neural representation learning.

where $n_t(z)$ and $n_f(z)$ are the number of preference rules that are true and false for $z$ respectively.

**Two-Step Learning Procedure.** We solve the relaxed optimization problem in eq. (1) in a two-step process. First, we learn a relational probabilistic decision tree (Blockeel and Raedt 1998) based solely on the structured portion ($Z$) of the data to predict the target label ($Y$). Each path from root to leaf in a decision tree can be viewed as a Horn clause. This tree-based partitioning divides the data space into related groups, each characterized by a conjunction of logical predicates. Let this tree be $\tau = (r, w)$ where $r$ is a function mapping the structured part ($Z$) of each data point to a leaf ($r(Z)$) in the decision tree and $w$ is a function mapping each leaf ($l$) to its potential value ($w(l)$). Then, relational probabilistic decision tree $\tau$ induces the conditional distribution:

$$P(Y \mid Z; \tau) = \sigma(w(r(Z)))$$

where $\sigma$ is the sigmoid function. This symbolic model provides a high-level interpretation of the relationship between structured data and the target and is learned by solving the following optimization problem:

$$\arg\min_\tau -\mathcal{L}(\tau, \mathcal{D}) - \lambda \sum_{(x,z,y) \in \mathcal{D}} w(r(z))(n_t(z) - n_f(z)).$$

In the second step, we refine the predictions made by the relational decision tree by incorporating information from the unstructured data ($X$). To do so, we train a neural network that takes as input both the raw unstructured data and an embedding representing the decision tree's prediction for a given instance. We consider a simple embedding consisting of the one-hot encoding of the leaf ID corresponding to the instance ($r(Z)$). Specifically, we learn a neural network $f$ to map this structured data embedding and the unstructured data to the functional newton-raphson update for the relational probabilistic decision tree. This value for an example $(x, y, z)$ and a relational decision tree $\tau = (r, w)$ is defined as $-g(z,y)/h(z,y)$ where $g(z,y) = P(y \mid z; \tau) - y - \lambda(n_t(z) - n_f(z))$ and $h(z,y) = P(y \mid z; \tau)(1 - P(y \mid z; \tau))$

are the functional gradient and hessian respectively. The overall output of the model is defined as

$$P(Y \mid X, Z; \tau, f) = \sigma(w(r(Z)) + f(X, r(Z)))$$

This allows us to use the expressive power of neural networks to capture complex interactions between the structured and unstructured data while grounding the learning process in the symbolic reasoning of the decision tree.

## Experimental Evaluation

We experimentally validated the ability of Deep Relational Dependency Networks (Deep-RDN) to effectively integrate structured (relational) and unstructured data on two datasets of varying complexity - ADE20K (Zhou et al. 2017) and RelKP (Wüst et al. 2024), which we describe below.

**Datasets.** ADE20K contains 20,000 scenes annotated with pixel-level segmentation maps for various objects present in the scenes. From this dataset, we extracted relational features such as object presence, spatial configurations, object sizes, and hierarchical relationships derived from WordNet to enable higher-level reasoning. The RelKP dataset consists of 200 Kandinsky patterns designed using relational clauses, representing varying levels of complexity based on the number of objects, object pairs, concept types, and relations. Examples of relations in this dataset include concepts such as "same shape" and "one object is a red triangle." ADE20K is thus a challenging dataset that allows the evaluation of reasoning under noisy real-world conditions, while RelKP on the other hand, is a controlled dataset that facilitates reasoning over well-defined relational structures. For our experiments, we focused on a subset of ADE20K containing 256 images and relations associated with the *Street* and *Highway* classes. For RelKP, we constructed a dataset with 50 images, corresponding to the complex concept "same color OR one is a red triangle." These datasets enable a thorough examination of our framework's ability to reason over both unstructured and structured data sources. To simulate noisy data, we randomly remove 15% of the relational predicates in the structured part and relabel 15% of the positive training examples as negative examples.

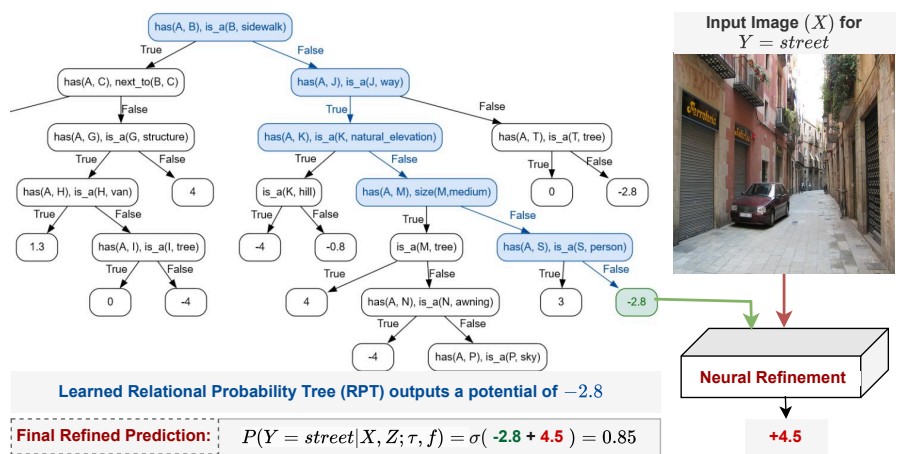

**Figure 2: Qualitative Illustration.** For a given input image corresponding to class=street, the relational probability tree (RPT), using only structured predicates, assigns a potential of $-2.8$, reflecting low confidence for the presence of a street, due to noisy or incomplete relational features. The neural refinement module uses unstructured visual cues to add a corrective potential of $+4.5$, increasing the total potential to $1.7$ and the final conditional probability to $0.85$. This demonstrates the effectiveness of combining relational reasoning with deep learning.

| Model | RelKP | | ADE20k-Highway | | ADE20k-Street | |
|---|---|---|---|---|---|---|
| | AUC-PR | AUC-ROC | AUC-PR | AUC-ROC | AUC-PR | AUC-ROC |
| NN | $0.86 \pm 0.09$ | $0.81 \pm 0.11$ | $0.88 \pm 0.04$ | $0.94 \pm 0.04$ | $0.85 \pm 0.08$ | $0.66 \pm 0.19$ |
| RPT | $0.85 \pm 0.12$ | $0.88 \pm 0.09$ | $0.73 \pm 0.14$ | $0.93 \pm 0.06$ | $0.91 \pm 0.03$ | $0.77 \pm 0.05$ |
| **Deep-RDN (Ours)** | $\mathbf{0.91 \pm 0.10}$ | $\mathbf{0.88 \pm 0.12}$ | $\mathbf{0.91 \pm 0.07}$ | $\mathbf{0.96 \pm 0.05}$ | $\mathbf{0.92 \pm 0.04}$ | $\mathbf{0.78 \pm 0.13}$ |

Table 1: **Performance Comparison** (mean $\pm$ standard deviation) of the three approaches—Neural Network (NN), Relational Probability Tree (RPT), and Deep Relational Dependency Network (Deep-RDN)—across the RelKP, ADE20k-Highway, and ADE20k-Street datasets in terms of Area Under the Precision-Recall Curve (AUC-PR) and Area Under the Receiver Operating Characteristic Curve (AUC-ROC).

**Methods.** We compare our framework against two baseline approaches: (1) a neural network classifier (NN) that only uses unstructured modality $X$, and (2) a relational probabilistic decision tree (RPT) learned from the structured modality ($Z$) and domain knowledge in the form of preference rules ($\mathcal{K}$). In contrast, our Deep-RDN model combines RPT with NN to use both unstructured ($X$) and structured ($Z$) data representations together with domain knowledge ($\mathcal{K}$). Since the unstructured modalities in both datasets are images, we used ResNet18 to encode each image as 512-dimensional vectors. All neural networks have one hidden layer of size 1/4th of the input dimension (512 for baseline NN, 512 + number of leaves in Deep-RDN). The neural networks for the Deep-RDN case were initialized by training them using the maximum likelihood objective function.

**Metrics.** We measure performance using two standard metrics: the Area Under the Receiver Operating Characteristic Curve (AUC-ROC) and the Area Under the Precision-Recall Curve (AUC-PR). AUC-ROC assesses the model's overall ability to discriminate between positive and negative examples at all possible thresholds, while AUC-PR evaluates the model's precision and recall across various thresholds, offering a more informative assessment in cases where class imbalance is significant.

**Results.** Table 1 presents the mean test AUC-PR and AUC-ROC scores for the three methods averaged over five folds of the RelKP, ADE20k-highway, and ADE20k-street

datasets. We observe that our Deep-RDN approach consistently outperforms both the NN-only and RPT-only baselines, demonstrating the value of jointly modeling structured relational knowledge and complex unstructured features. Notably, for the ADE20k-Street subset, Deep-RDN achieves approximately a 7% improvement in AUC-PR and a 12% improvement in AUC-ROC compared to the NN baseline, underscoring the benefits of combining structured knowledge with learned visual features.

## Conclusion

In this work, we introduced Deep Relational Dependency Networks (Deep-RDN), a novel framework for learning relational models by effectively integrating structured and unstructured data with domain knowledge. The proposed model combines the interpretability and reasoning capabilities of probabilistic relational logic with the rich representation learning capabilities of deep neural networks. Our experimental results highlighted the advantages of this hybrid framework in scenarios with noisy and incomplete data. Future research directions to enhance the framework's adaptability include extending it to support multi-class and multi-label classification tasks, scaling it to accommodate larger multimodal datasets with more complex relational structures, and enabling end-to-end learning where the symbolic component and neural module iteratively exchange feedback to improve each other's learning and performance.

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
