# OpenReview forum: "Learning Probabilistic Logic Models over Structured and Unstructured Data"
_AAAI.org/2025/Workshop/NeurMAD — AAAI 2025 Workshop NeurMAD Submission_

### Official Review · Reviewer_T8rg · 2024-12-19
**Reviews for Sub23**

**Rating:** 4
**Confidence:** 4

**Review:**

1.It's better to do comparison with the state-of-the-art decision deep networks/tree-based models
2.it's better to do experiments for large-scale datasets

---

### Official Review · Reviewer_7hHF · 2024-12-20
**A new approach towards neuro-symbolic integration**

**Rating:** 7
**Confidence:** 4

**Review:**

This paper proposes a new method to combine relational dependency networks (in the form of relational probabilistic decision trees (RDTs)) with neural networks in order to leverage both structured and unstructured data for decision-making in noisy environments. Overall, the paper is well-motivated, and the methods are clearly described. While only a short paper, it might still benefit from more elaborate experiments that also compare to the state-of-the-art methods.

Some open questions/limitations that could be addressed
* Why would the logical penalty function (end of page 2) include phi that depends on unstructured data? This way, the unstructured data could invalidate the logical penalty function. Indeed, the penalty function on page 3 doesn’t contain any contribution from the unstructured data anymore.
* It is unclear how we can balance the contributions of neural refinement and the RPT.
* It would be interesting to see how the model behaves with different levels of noise in the structured data as well as the labels. I assume that with zero noise, an RPT would be sufficient to achieve high accuracy. When (at which noise level) does it become helpful to consider a combination of RPT and NN?
* Where do we get the labels for the RPT? Could this be provided by another NN?

---

### Official Review · Reviewer_Znpw · 2024-12-25
**Novel Approach but Choice of Dataset (Highly Unstructured only) not suited for Experiments done**

**Rating:** 5
**Confidence:** 4

**Review:**

## Summary
---
This paper presents Deep Relational Dependency Networks (Deep-RDN), a framework that combines relational dependency networks and neural networks to integrate structured and unstructured data. It employs a decision tree for structured data and a neural network for unstructured inputs, incorporating domain knowledge through preference rules. Tested on ADE20k and RelKP, the model outperforms baselines in noisy, multimodal tasks.

## Strengths
---
- Novel combination of symbolic reasoning and deep learning with consistent improvements over baselines, especially in noisy scenarios.
- Comprehensive evaluation with clear metrics and integration of domain knowledge for enhanced interpretability.

## Suggestions for Improvement
---
- Dataset Choice: ADE20k is primarily unstructured and doesn’t align well with healthcare or other high-stakes applications. Medical datasets like MIMIC-IV or CheXpert, which combine structured records and unstructured imaging/text, would better validate the model's utility.

- Missed Citations: The paper overlooks related work in neurosymbolic reasoning and multimodal learning frameworks, such as approaches integrating neural networks with probabilistic graphical models (e.g., Learning using Privileged Information by Vapnik, hybrid neurosymbolic frameworks).

While innovative, addressing these gaps and testing on more relevant datasets would enhance the paper's alignment with its stated high-stakes application focus.

---

### Decision · Program_Chairs · 2024-12-30

**Decision:**

Reject

**Comment:**

 This paper does not fit the scope of this workshop. It has some novelty but a bit old-fashioned by targeting at learning probabilistic logic models.